# Opinion Polls and Antibody Response Dynamics of Vaccination with COVID-19 Booster Vaccines

**DOI:** 10.3390/vaccines10050647

**Published:** 2022-04-20

**Authors:** Yufei Wu, Huanjie Li, Yangyang Wang, Ping Huang, Yihui Xu, Mingjie Xu, Qianqian Zhao, Yunying Zhou, Jun Wang, Mingyu Ji, Yunshan Wang

**Affiliations:** 1School of Medicine, Cheeloo College of Medicine, Shandong University, 44 West Wenhua Road, Jinan 250012, China; gai1027@126.com (Y.W.); wangyangyang89757@163.com (Y.W.); labmedzxyy@163.com (M.J.); 2Medical Research and Laboratory Diagnostic Center, Jinan Central Hospital Affiliated Shandong First Medical University, 105 Jiefang Road, Jinan 250013, China; huangping727@163.com (P.H.); xuyihuih@163.com (Y.X.); xmjmjj1984@163.com (M.X.); qqzhao1023@hotmail.com (Q.Z.); joan0539@163.com (Y.Z.); junwang0523@163.com (J.W.); 3Department of Epidemiology, School of Public Health, Cheeloo College of Medicine, Shandong University, 44 West Wenhua Road, Jinan 250012, China

**Keywords:** COVID-19, SARS-CoV-2, booster vaccines, antibody response dynamics

## Abstract

As the third year of the global COVID-19 pandemic, vaccination remains the most effective tool against infections and symptomatic illness. Comprehension regarding immunity to SARS-CoV-2 is limited, and the durability of immune responses after vaccination is currently not clear. In this study, we randomly collected 395 questionnaires to analyze the current state of COVID-19 vaccination. At the same time, the serum of 16 individuals who had received two doses of the COVID-19 vaccine were collected at different times before and after the booster vaccination. We analyzed the dynamic changes of SARS-CoV-2 S-specific binding antibodies in serum and immunological indicators. By collecting public opinion surveys and analyzing variational trends of SARS-CoV-2 S-specific binding antibodies and immune indicators after COVID-19 booster vaccination, we endeavored to demonstrate the concerns affecting people’s booster vaccinations, as well as the frequency, timing, and necessity of COVID-19 booster vaccinations. The analysis of antibody results in 16 vaccinated volunteers showed that the antibody concentration decreased six months after the second dose and the protective effect of the virus was reduced. The third dose of COVID-19 vaccination is necessary to maintain the antibody concentration and the protective effect of the virus. The vaccination with the vaccine booster depends not only on the time interval but also on the initial concentration of the SARS-CoV-2 S-specific binding antibody before the booster. Our study has important implications for raising public awareness of vaccinating against SARS-CoV-2 and the necessity of COVID-19 booster vaccinations.

## 1. Introduction

COVID-19 has been a global pandemic that has caused more than 350 million infections and five million deaths worldwide. There has been no sign of reduction since 29 December 2019 [1,2]. Billions of doses of severe acute respiratory syndrome coronavirus 2 (SARS-CoV-2) vaccines have been given around the world since the first vaccine was approved [3]. SARS-CoV-2 has continued to mutate in the intervening years. Despite the clinical diagnosis and the therapeutic measures that have been achieved, the antibody response is excellent in the general population after two doses of vaccine against SARS-CoV-2 [4]. The Omicron variant may escape vaccine-induced immune protection more easily. After two doses of inactivated vaccine, a third heterologous protein subunit vaccine or a homologous inactivated vaccine booster improved Omicron neutralization [5]. Therefore, there is a necessity to require the development of vaccines that provide long-lasting immunity [6]. Vaccines represent one of the most efficacious biomedical methods to improve society by saving lives and dramatically decreasing the spread of infectious disease [7].

Despite the importance of vaccinology, further research is still needed to understand how the best COVID-19 vaccines work to achieve better protective efficacy. Previous studies have shown that infection with the measles virus results in stable serum antibody responses which are largely maintained above a protective threshold for life [8]. Indeed, the effectiveness measurements of immunity from vaccines are impacted by these variations, for example, natural infection, population behavior, and public health policies. The immunity will wane over time [9,10]. This decrease in immunity was confirmed in several observational studies reporting a decrease in effectiveness against infection five to six months after the second dose [11]. Therefore, the third dose booster vaccine has been proposed with a vaccinal interval requiements which is more than 6 months, but remains questionable.

COVID-19 is often associated with immunopathology levels of T lymphocytes, including regulatory T cells and cytotoxic and helper T cells [12,13]. Patients with severe illness showed a cytokine storm syndrome associated with dysregulated immune activation and hyperinflammation [14]. Humans make SARS-CoV-2-specific antibodies, CD4+ T cells, and CD8+ T cells in response to SARS-CoV-2 infection [15]. Studies of acute and convalescent COVID-19 patients have observed that T cell responses are associated with reduced disease [16].

Although natural infection will often elicit lifelong immunity, almost all current vaccines require a booster vaccination in order to achieve durable protective humoral immune responses, regardless of whether the vaccine is based on infection with replicating live-attenuated vaccine strains of the specific pathogen or whether they are derived from immunization with inactivated, non-replicating vaccines or subunit vaccines [17]. The form of the vaccine antigen (e.g., soluble or particulate/aggregate) appears to play an essential role in determining immunogenicity and the interactions between dendritic cells, B cells and T cells in the germinal center are likely to dictate the magnitude and duration of protective immunity [18,19]. Circulating SARS-CoV-2-specific T cell responses correlate poorly with late immune responses that control viral load, which is not surprising since these cells need to migrate from the circulation to other locations, such as the respiratory tract, to control the virus [20].

Currently, the inactivated vaccine requires two doses on days 0 and 14 (or 0 and 21, 0 and 28). Inactivated vaccine shows ideal protection 14 days after the second dose. The inactivated vaccine showed ideal protection around 14 days after the second dose [21]. Neutralizing titers in convalescent sera of COVID-19 patients decreased significantly at six months [22]. It is unclear whether a similar situation will occur with the inactivated vaccine at some point after two doses of the vaccine. Moreover, the necessity of improving the efficacy and persistence of inactivated vaccines by booster injections needs to be further explored. Considering individual differences, can we consider changing the time point for of the COVID-19 booster vaccines? When the level of neutralizing antibodies in the body decreases, should booster injection be carried out to maintain the potency of the vaccine?

In this study, we randomly selected 395 individuals including healthy adults between age 18 to 65 and adolescents between the ages of 12 to 18 to analyze the polls and the current state of COVID-19 vaccination. At the same time, in order to assess the safety, reactogenicity and immunogenicity of the vaccine, we randomly selected 16 individuals who had received two doses of the COVID-19 vaccine collect their serum seven days before they received a booster vaccine and 7, 14, 21, 28 and 56 days after vaccination. By analyzing public opinion surveys and variation trends in SARS-CoV-2 S-specific binding antibodies and immune indicators of COVID-19 booster vaccination, we are committed to demonstrating the concerns affecting people’s booster vaccinations, as well as the frequency, timing, and necessity of COVID-19 booster vaccinations.

## 2. Materials and Methods

### 2.1. Trial Design and Participants

The trial was divided into two parts. The first part was an opinion survey in the form of online questionnaire survey by WeChat Questionnaire Star software. We conducted an opinion survey in 395 healthy adults between the ages of 18 and 65 years and healthy adolescents between 12 and 18 years. The second part involved clinical trials. We recruited 16 volunteers who had received two doses of COVID-19 inactivated virus vaccine (Sinopharm Beijing, China) and were willing to receive booster shots. The trial design called for an evaluation of the boosting effect of the third dose of COVID-19 inactivated virus vaccine at seven days, 14 days, 21 days 28 days and 56 days after vaccination with respect to safety, reactogenicity, and immunogenicity in each volunteer. The study was conducted according to the guidelines of the Declaration of Helsinki and approved by the Institutional Review Board of the Ethics Committee of Jinan Central Hospital (Ethical code No. D202111Ab). Both the questionnaire and the informed consent form are available. All of the participants provided written informed consent.

### 2.2. Opinion Polls Design

The questionnaires investigated participants’ vaccine attitudes by asking about getting vaccinated against COVID-19. They were divided into three parts which contained basic characteristics around cognitive consciousness of COVID-19 vaccines and vaccination situations. The data reported by participants as e part of a self-completed questionnaire were collected from questionnaire offered by WeChat mini program Questionnaire Star included the participants’ sociodemographic characteristics, medical history, vaccination uptake, the main reasons they chose to be vaccinated or not, their knowledge about immunoreaction after vaccination, and the influence of the COVID-19 pandemic on vaccination. Each section of the questionnaire included a set of items, and the respondents were asked to choose a predefined answer listed after a question or statement. In addition, an information sheet explaining the study was supplied (Appendix A).

### 2.3. Specimen Collection

The samples were collected from 16 vaccination volunteers who had received COVID-19 booster vaccines. Nasopharyngeal and oropharyngeal swabs of the volunteers were inserted into one sterile tube containing 3 mL of virus preservation solution. They also provided 2 mL of blood and 2 mL of serum by venous blood collection from seven days before vaccination and seven, 14, 21, 28, and 56 days after vaccination.

### 2.4. Laboratory Analysis

The SARS-CoV-2 virus RNA was extracted using the magnetic beads method, according to the instructions of the nucleic acid extraction kit (Shanghai Zhijiang Biotechnology Co., Ltd., Shanghai, China) and then tested by RT-PCR following the steps of the kit in a tertiary protection laboratory. The collected blood samples were centrifuged at 3000 rpm for 10 min to isolate serum and stored at 4 °C before future experiments. All the experiments were carried out at the BSL-2 laboratory. We used the magnetism particulate chemistry method luminescence assay (CMIA) to measure SARS-CoV-2 S-specific binding antibodies at baseline seven days before vaccination and on days seven, 14, 21, 28, and 56 after vaccination with aWanTai 2019-nCoV antibody detection kit. The kit adopts a recombinant sample and 2019-nCoV coating antigen to measure the chemical immunofluorescent signal. The results were showed by RLU which were positively relative to the contents of SARS-CoV-2 S-specific binding antibodies in the serum. The luminometer measured the relative luminescence value RLU average of calibrator 1 and calibrator 2 three times, and calculated the cut-off index (COI). When the samples test value < 1.0 (COI), the 2019-nCoV antibody test result was judged to be negative. When the samples test value ≥ 1.0, the 2019-nCoV antibody test result was determined to be positive. We used intracellular cytokine staining, and T-cell responses were measured at baseline and seven days before vaccination and seven, 14, 21, 28, and 56 days after vaccination. In T cells, we achieved a response in CD4, CD8, CD69 and CD25. In CD19 cells, a response in CD69 and CD25 was realized by flow cytometry detection kit (4ABIO Tech. Co. Suzhou, China) of CD3, CD4, CD8, CD25 and CD69 according to the manufacturer’s protocol.

### 2.5. Statistical Analysis

For descriptive statistics, frequencies and percentages were calculated for each question. Linear regression was performed between each question and vaccination intentions. The relationship was considered significant if *p* < 0.05 or *p* < 0.01. For blood sample data, we used log-transformed data to calculate the confidence intervals of the geometric means, and used Spearman’s correlation analysis between times interval with antibody concentration. All analyses were performed with SPSSAU.

## 3. Results

### 3.1. Research Process Flow

Opinion poll data was collected from the questionnaire offered by the We Chat mini program questionnaire star and 395 valid questionnaires were collected for further analysis. We recruited 16 volunteers whose nucleic acid tests were negative and had who had received two doses of the COVID-19 vaccine, and then they received the third dose of COVID-19 vaccine voluntarily. On 22 October 2021, the trial was initiated in Jinan Central Hospital in Shandong Province, China. Informed consent for the project was provided by all volunteers. Both the questionnaire and the informed consent form are available as a Appendix A. Volunteers received a 5 mL COVID-19 vaccine which was administered intramuscularly in a single-dose. The trial design was called to for an evaluate of the effect of booster shots at seven, 14, 21, 28 and 56 days after vaccination to safety, reactogenicity and immunogenicity in each volunteer from 22 October to 24 December. The whole course is shown in a process flow diagram in Figure 1.

### 3.2. Respondent Characteristics

The characteristics of 395 people were processed for analysis covering gender, age, education, occupation, health, basic illness and vaccination condition. In total, 138 (34.94%) persons received the third dose of inactivated COVID-19 vaccine (Details regarding all characteristics are provided in Table 1). “Healthy” means that the participants do not have any underlying diseases, have good immunity, and rarely catch colds, while “healthy” generally means that the participants have some underlying diseases that do not affect their normal life [23,24]. The sociodemographic characteristics of the 395 persons are shown in Table 1. More than half (56.96%) of the respondents were male and the majority (98.22%) were between 18 and 60 years of age. 80.76% of them had a high level of education (Bachelor’s degree or above), as more than a third of them had a Master’s degree or higher. In addition, 93.42% were employed (including 17.34% students). The majority (88.1%) of the respondents were in good physical condition, and the rest (11.9%) of them had basic illnesses, including CVD, chronic tumor, chronic respiratory diseases, immune deficiency diseases or other chronic diseases.

### 3.3. Estimation of Preferences and Heterogeneity

Among the 395 participants in the study cohort, 366 (92.66%) of them were willing to receive a third dose of the COVID-19 vaccine, and 138 of them (34.9%) had received the third dose of the vaccine. A linear regression was made between the willingness to vaccine booster shots with participants’ characteristics and cognitive levels (Appendix A). As presented, the results showed that there was no correlation between the willingness to receive vaccine booster shots and the participants’ characteristics. Awareness of COVID-19 is positively correlated with vaccination intention (*p* < 0.05). There was a negative correlation between the perception of temporary discomfort after vaccination and the willingness to receive a vaccination (*p* < 0.05). The positive correlation between awareness of COVID-19 vaccine precautions and the willingness to get vaccinated (*p* < 0.01) suggested that people have a certain understanding of virus prevention. Similarly, attention to COVID-19 vaccine research and development is positively correlated with vaccination intention (*p* < 0.01). People’s willingness to vaccinate is closely related to individual immune response after vaccination, the understanding of vaccine-related precautions, the attention to vaccine research and development, and the awareness of the harm of COVID-19. The individual immune response after vaccination, the understanding of vaccine-related precautions, the attention to vaccine research and development, and the awareness of the harm of COVID-19 are the main reasons that influence people’s willingness to get COVID-19 booster vaccines. In general, as presented in Figure 2, 44% of participants experienced a mild immune reaction after receiving the COVID-19 booster vaccine, and 84% of participants considered it to be of great importance to get a COVID-19 booster vaccine (Appendix A).

### 3.4. Dynamic Serosurvey of SARS-CoV-2 S-specific Binding Antibodies Concentration

In order to study the trend of SARS-CoV-2 S-specific binding antibodies before and after booster vaccination with the COVID-19 vaccine and the difference between the individuals, SARS-CoV-2 S-specific binding antibodies were detected in the serum of 16 volunteers who received booster shots. The SARS-CoV-2 S-specific binding antibodies were performed at seven days before booster vaccination and separately at seven, 14, 21, 28, and 56 days after vaccination (Figure 3).

We found that the level of S-specific binding antibodies was low, and the antibody titer value was between 0.8 and 20 (COI) at seven days before booster vaccination, which was close to the negative level (COI < 1.0). Antibody levels gradually increased after vaccination, reaching the highest level at 14 days after vaccination, and the antibody titers of several volunteers remained above 1000 (COI). From 14 days to 56 days, the specific antibody levels of the volunteers tended to stabilize (Figure 4a). As presented in Figure 4b, for personal trend analysis of SARS-CoV-2, S-specific binding antibodies were performed for seven days before booster vaccination and seven, 14, 21, 28, and 56 days after vaccination. Vaccinators 2, 6 and 14 indicated that the initial antibody concentration value was very low, and the peak rate of antibodies after booster injection was lower than that of other vaccinators. The initial antibody concentration of No.8, No. 13 and No. 15 vaccinees was higher, and the peak antibody rate after booster injection was higher than that of other vaccinees. The results showed that the peak antibody concentration of volunteer No. 2 was lower than that of the other volunteers. We explored the sources of heterogeneity. By analyzing the age composition of the volunteers, we found that volunteer No. 2 is 60 years old, which may be a potential source of heterogeneity. The antibody concentration value of volunteer 10 began to increase at 14 days, and did not increase as rapidly as other volunteers seven days after the booster injection. Therefore, we speculate that the initial concentration of antibodies and the time interval affect the peak antibody concentration of an individual after booster vaccination. When the initial antibody concentration was lower than 4 (COI), the antibody concentration did not have a tendency to increase after the third dose.

### 3.5. Correlation Analysis of SARS-CoV-2 S-specific Binding Antibodies Concentration

Through correlation analysis, we found that the peak of antibody concentration was related to several factors through the difference chart of individual antibody variation trends. A correlation analysis was made between the initial SARS-CoV-2 S-specific binding antibody, the time interval of vaccination and the antibody trend. A Spearman’s analysis showed that the antibody concentrations at 14, 21 and 28 days after booster vaccination was significantly correlated with concentrations at seven days before vaccination (*p* < 0.01) significant correlation at 56 days (*p* < 0.05) in Figure 5a. There was no significant correlation between the antibody concentration seven days after booster vaccination and seven days before vaccination in Figure 5a and Appendix A. As shown in Figure 5b, the concentration of SARS-CoV-2 S-specific binding antibodies 14 days after booster vaccination was positively correlated with the initial antibody concentration before vaccination. The concentration of SARS-CoV-2 S-specific binding antibodies after 21 days was also positively correlated with the initial antibody concentration in Figure 5c. The higher the initial concentration, the higher the concentration of SARS-CoV-2 S-specific binding antibody at 28 days after booster vaccination, as shown in Figure 5d. In Figure 5e, the higher the initial concentration, the concentration of SARS-CoV-2 S-specific binding antibody was higher at 56 days after inoculation. It can be clearly seen from Figure 5b–e that the concentration of SARS-CoV-2 S-specific binding antibodies after booster inoculation 14 days and 21 days. The trend that was positively correlated with the trend in initial antibody concentration before vaccination was more significant than the trend at 28 and 56 days. Those results suggested that the peak intensities of antibody concentration were related to the antibody concentration before booster vaccination.

Although the correlation between the concentration of SARS-CoV-2 S-specific binding antibody and the interval time after booster vaccination was not significant in Figure 5a, we still wondered what their relationship was. Conversely, the change of antibody at 14, 21, 28, or 56 days after vaccination were shown towards a different interval. The longer the interval between the second and third injection, the lower the antibody concentration was (Appendix A). When the interval of time is more than eight months, the change range of antibody resistance concentration is lower than the interval at six to eight months (Appendix A). Those results suggested that it is better to accept the booster vaccine at six months. Most importantly, the antibody level should not be lower than 4 (COI).

### 3.6. CD4+ T Cell and CD8+ T Cell Responses before and after the Third Vaccination

B cells, CD8+ T cells, and CD4+ T cells, have relative immune memory kinetics, and understanding the immune memory of vaccine effects may require the evaluation of its various components. By grasping the complexities of SARS-CoV-2 immune memory, it may be key to gain insight into protective immunity to SARS-CoV-2 reinfection and secondary COVID-19 persistence [25]. In the current study, we evaluated CD4+ T and CD8+ T cell responses before and after the third dose of the COVID-19 vaccine and assessed the safety of vaccine booster shots at different times (Appendix A). CD4+ T cells rose on the 15th day after vaccine injection and decreased at the 28th day, as shown in in Figure 6a. The result of CD8+ T cell showed an opposite trend with CD4+, decreasing at 15 days and increasing at 28 days in Figure 6b. The ratio of CD4+/CD8+ has now been used as a clinically conventional index to evaluate TB patients’ immunity [26]. We analyzed CD4+/CD8+ to understand the state of immune function after the third booster dose of the vaccine. After booster vaccination, the ratio of CD4+/CD8+ showed a modest increase until day 21 and then decreased to 1.0 and tended to be stable, which was a little higher than before, as shown in Figure 6c,d. Those results indicate that there will be a period of immune response after vaccination showed an increasing immune level, and the immunity will be enhanced in the later stage. The surface antibody expression of CD25 and CD69 was also detected. The results showed that CD25 + CD4+ T cells increased significantly on day 56. CD69 + CD4+ T cells increased on day 56, but the trend was not significant. CD25 + CD69+ in CD8+ T cells showed a downward trend and increased slightly at 56 days (Appendix A).

## 4. Discussion

After the COVID-19 outbreak, vaccination has become one of the most effective ways to prevent the broadest spread of the virus. The inactivated vaccine can cause a series of immune responses after entering into the human body, and understanding the characteristics of the immune response is of great significance in the development and application of effective vaccines. However, the profiles of the immune system after vaccination are currently not fully understood. This study conducted a questionnaire survey to investigate the willingness of people to receive booster shots of the COVID-19 vaccine and the factors affecting people’s vaccination choices. In addition, we attempted to explore the relationship between the alterations in the immune system including the SARS-CoV-2 S-specific binding antibodies and T cells in 16 volunteers before and after receiving a COVID-19 vaccine booster shot to find a relatively more appropriate timing of the vaccination. Despite the fact that getting a vaccine is unquestionably important, it is important to study the role of risk perception in responses and individual immunity levels to preventive behavior.

The antibody levels increased greatly among the 16 volunteers who reported receiving a COVID-19 booster vaccination. Therefore, we speculate that the COVID-19 immunity level may increase after the booster vaccination, which further clarifies the necessity of the booster vaccination. At present, the mortality rate of patients infected with COVID-19 in China has decreased significantly. On the one hand, this means that the public’s awareness of prevention has improved. On the other hand, it is closely related to the widespread vaccination of the public.

The results of our questionnaire showed that the willingness and intention of people to get the booster shot were related to the immune response caused by vaccines, and the immune response caused by vaccines became one of the important reasons why people were unwilling to take them. Risk perception depends on the specific characteristics of the hazard. In the case of COVID-19, it has become a close physical and psychological threat during the transition from pre-lockdown to lockdown. Although understood poorly in the scientific community, therapeutic intervention is limited and the course is unpredictable [27]. A recent survey shows that educating older adults about the safety of the COVID-19 vaccine will help them make decisions [28]. By now, mRNA vaccines are not available to most people in the world (excluding China) due to an insufficient number of vaccines being available. Considering the lack of pragmatic evaluation of the effectiveness on collective health indicators [29], r studies have shown that the Omicron variant reduces neutralizing antibody levels by 10–40-fold and avoids binding to most, but not all, monoclonal antibodies that directly interfere with the interaction of ACE2 with the S protein [30]. Some recent surveys have found that the most important factor influencing vaccine preference is vaccine efficacy, while negative effects reduce the likelihood of vaccine acceptance [31]. They are also the reasons why people have doubts about getting vaccinated. Because of the skeptical concern about the booster vaccination, it is necessary to strengthen public knowledge about vaccine science and safety in order to make more people decide to get the booster vaccination. Therefore, the government should strengthen the publicity and knowledge of vaccination to improve the people’s trust and increase vaccination coverage.

The analysis of antibody results in 16 vaccinated volunteers showed that the antibody concentration decreased six months after the second dose and the protective effect of the virus may have been reduced. These results suggest that the third dose of COVID-19 vaccination is necessary to maintain the antibody concentration and the protective effect of the virus. In the present study, the SARS-CoV-2 S-specific binding antibody level raised gradually after vaccination, and the highest level appeared at 14 days after vaccination and then stabilized. After the CD4+ T cells level was detected, we found that the level of CD4+ began to rise on the 15th day of vaccine injection and decreased on the 28th day. The expression of surface antibodies of CD8+ T cells showed an opposite trend with CD4+. It is still unknown what levels of antibody are appropriate to protect the people from severe symptomatic disease or infection with SARS-CoV-2. Additionally, previous studies did not consider the impact of vaccine generated memory cells on the duration of protective immunity. In this study, we provided further evidence that a booster vaccination may improve an individual’s anti-infective performance against the virus. In addition, we emphasized the necessity of improving the effectiveness and durability of inactivated vaccines through booster shots, although some previous studies reported that immune memory was still active at six months after the second dose [25]. More studies are required to confirm this finding and fully elucidate the underlying immunological principles.

Our study also found that the administration of a COVID-19 vaccine booster might not only be related to the time interval, but also be more significantly related to the initial concentration of antibodies before the booster. The above considerations suggest that the individual booster doses should not be defined by a specific time, but that vaccine recipients should be monitored for changes in the decline in antibody concentrations after the second dose. The initial concentration of SARS-CoV-2 S-specific binding antibody has a direct effect on the peak of antibody concentration after inoculation. When the initial concentration was below 4 (COI), the SARS-CoV-2 S-specific binding antibody concentration would not rise to a high level after the third dose. Therefore, we recommend that when the SARS-CoV-2 S-specific binding antibody concentration is lower than 4 (COI), the general public should promptly receive the third booster vaccine. Dynamics within the host are as important as the contact structure in determining transmission: the most vulnerable individuals not only have a higher chance of becoming infected, but also of dying from infection. In addition, with high viral loads concentrated in elderly residential areas and hospitals, this segment is not only more vulnerable, but more contagious [32]. Due to some individual heterogeneity, antibody levels in elderly vaccinated subjects decrease rapidly and increase slowly. The time point of the booster shot of the COVID-19 vaccine should be based on SARS-CoV-2 S-specific binding antibody concentration. We speculate that a fourth shot will be needed before finding a specific drug for COVID-19 and developing a more effective vaccine.

CD25 + CD4+ T cells are regulatory T cells, Tregs, and cytokines that suppress the immune response and suppress cytokine storms in infectious diseases [33], so we detected the surface antibody expression of CD25 and CD69. Although monovalent protein antigens will elicit better and more durable antibody responses (due to the acquisition of T cell help), the most long-lived antibody responses are predicted to occur when a multivalent triggers strong B cell activation as well as effective T cell help [7]. CD69 is a marker of the early activation of T cells, and very few T cells express CD69 on the surface, which shows enhanced immunosuppressive ability after expression [34]. After activation of CD69, it can promote the expression of CD25 as a signal of the early activation of T cells and stimulate the expression of CD4+ T cells.

Taken together, we found that the virus tends to persist for a long time in humoral immunity, but antibody levels decline more rapidly for those without repeated immunity. Over time, after the second injection, the mortality rate and severe disease rate decreased significantly even if some people became infected with the virus. Vaccines against emerging variants are less effective, so vaccines need to be adapted to work specifically against variants and to be rolled out quickly. Booster immunization programs that may provide better immune responses and protection need to be developed across vaccine platforms [35]. People who are recovering from COVID-19 should also be vaccinated, and those who have been vaccinated should receive boosters to boost their immunity to emerging variants such as Omicron [36]. Several reports have also demonstrated that booster vaccines induce significant increases in neutralizing antibody titers against the wild-type SARS-CoV-2 virus as well as the beta and delta variants [37].

Human coronaviruses are primarily transmitted by respiratory droplets, but aerosol, direct contact with contaminated surfaces, and fecal-oral transmission has also been reported during the SARS epidemic [38]. Booster vaccines alone are not enough to solve the COVID-19 epidemic. A recent report found that one possibility for SARS-CoV-2 is to target autophagy, which may be responsible for the ineffective immune response to uncontrollable viral replication and thus to the end of the spiral disease [39]. If we can cut off the transmission route, inject vaccines, develop new drugs and treatments based on the molecular transmission mechanism of SARS-CoV-2 and combine these methods, the COVID-19 epidemic will be resolved in a more rapid manner.

Data on the duration of vaccine protection and the realistic effects and safety of booster doses are the basis for increasing the willingness of the population to get vaccinated. While booster shots are important, vaccination regimens should give priority to the individuals who have not been vaccinated to date.

## 5. Conclusions

In short, this study has important implications for addressing the COVID-19 pandemic and raising public awareness of vaccination. From our study, it is concluded that researchers need to develop new COVID-19 vaccines against the new mutant strains. We speculate that there are more than three booster shots required, and when the antibody level drops to a certain level, a fourth booster shot should be given. Booster vaccination should not be limited to six months after receiving a vaccine, and we speculate that the timing, frequency and dose of booster vaccination should be considered according to individual differences. When the antibody concentration in the body is lower than 20 (COI), a booster shot should be administered quickly.

## 6. Limitations

First, the 16 volunteers tended to be young people with postgraduate educations who may have had some deviation in their understanding of vaccines. Second, the results were restricted to a population group with a high vaccination rate that represents the majority of the group. Additionally, the analyses of COVID-19 vaccination in this study are only based on the inactivated vaccine. Finally, it should also be noted that a significant number of people received the mRNA vaccine, so the results should be interpreted with caution.

## Figures and Tables

**Figure 1 vaccines-10-00647-f001:**
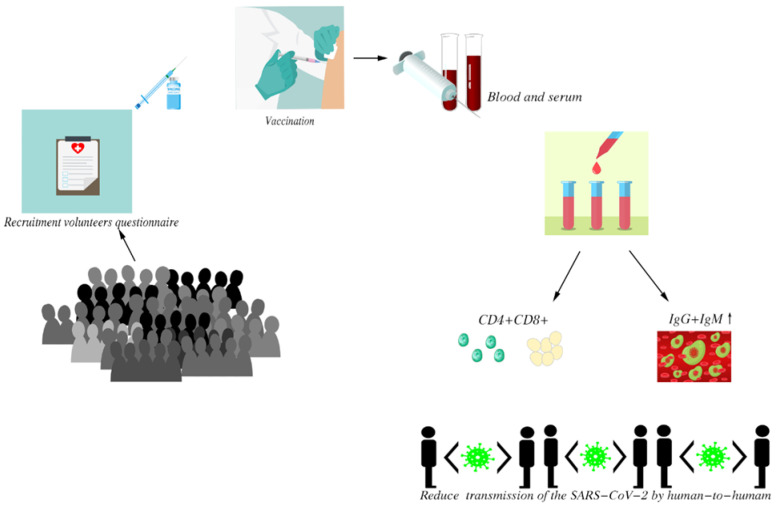
Research process flow diagram.

**Figure 2 vaccines-10-00647-f002:**
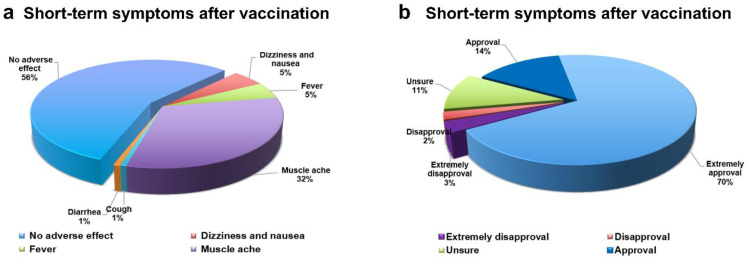
Percentage of individual symptoms and vaccination trust rank survey. (**a**) Reference = 34.94%: 34.94% of survey participants had been vaccinated with booster shots; all percentages are for vaccinated booster participants. (**b**) The pie chart represents a vaccination scale for the 395 questionnaire respondents divided into 5 sections.

**Figure 3 vaccines-10-00647-f003:**
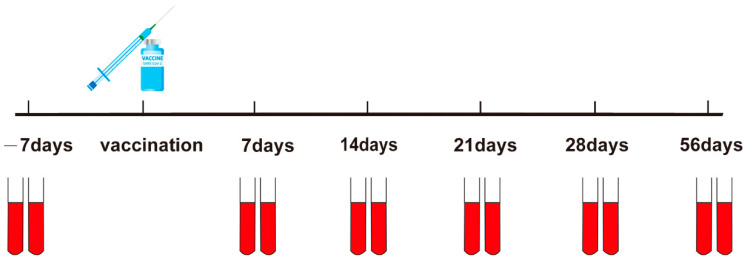
Time axes of samples collection and inoculation. −seven days, the seven days before the COVID-19 booster; seven days, 14 days, 21 days, 28 days, and 56 days, The seven days, 14 days, 21 days, 28 days, and 56 days after the COVID-19 booster vaccine.

**Figure 4 vaccines-10-00647-f004:**
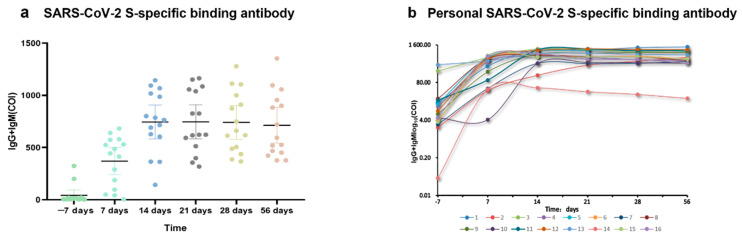
Antibody responses specific to SARS-CoV-2 vaccination in 16 volunteers. (**a**) The scatter chart is the antibody trend chart of the six time points before and after vaccination. I Bars indicate 95% confidence intervals. (**b**) The line chart shows the trend of antibody changes in 16 volunteers at six time points before and after vaccination, respectively; *Y*-axis values are log_10_ of antibody titer values.

**Figure 5 vaccines-10-00647-f005:**
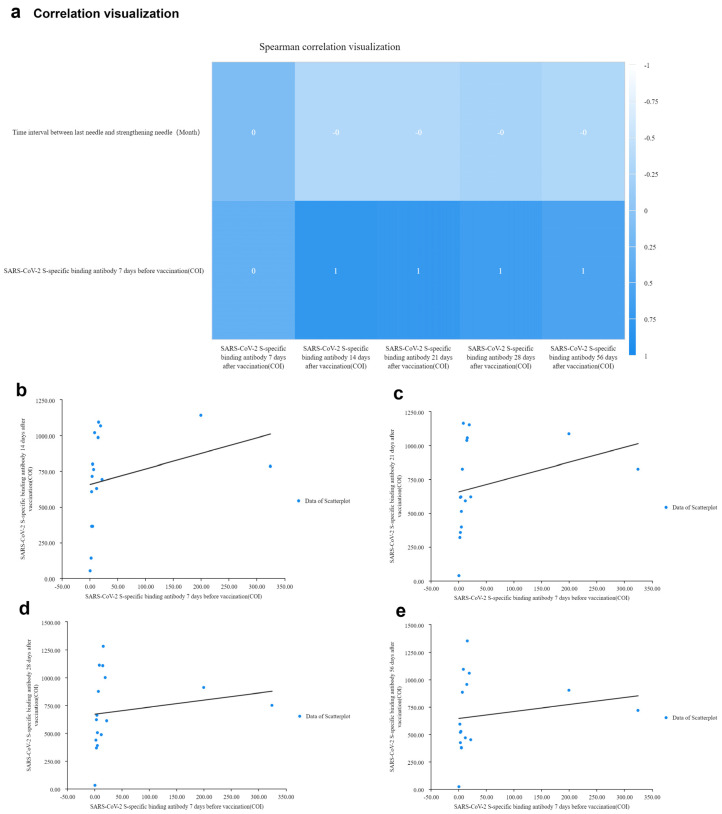
Correlation analysis of SARS-CoV-2 S-specific binding antibodies concentration. (**a**) Correlation analysis between SARS-CoV-2 S-specific binding antibody concentration and booster vaccination interval; correlation analysis between SARS-CoV-2 S-specific binding antibody concentration and initial antibody concentration. The higher the correlation between the two quantities, the closer the *p*-value is to 1, the darker the blue. (**b**–**e**) Trend graph of the correlation between initial antibody concentration and antibody concentration at 14 days, 21 days, 28 days, and 56 days after vaccination.

**Figure 6 vaccines-10-00647-f006:**
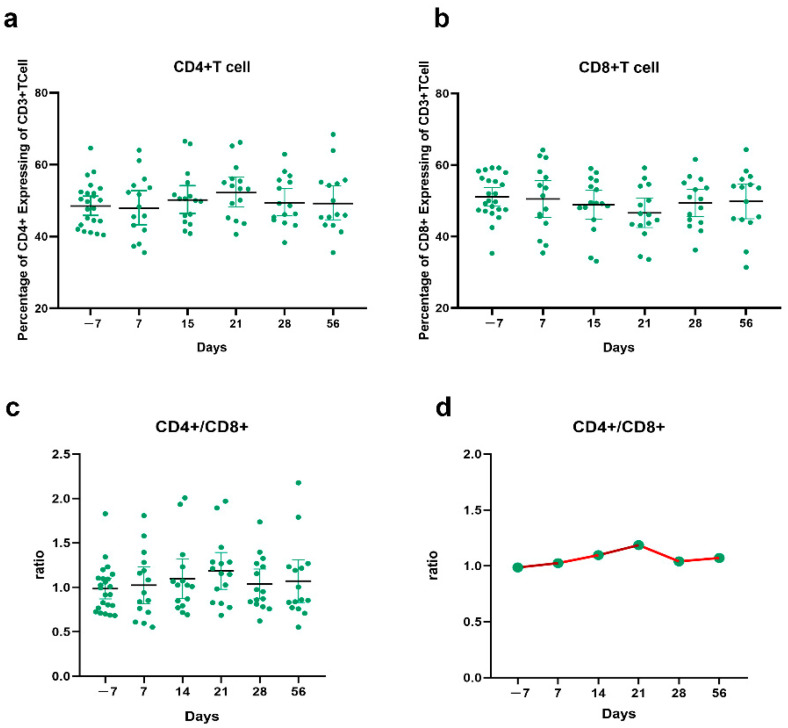
The trend of CD4+, CD8+ T cells responses to SARS-CoV-2 vaccination. (**a**) A CD4+ trend chart of the six time points before and after vaccination; (**b**). A CD8+ trend chart of the six time points before and after vaccination; (**c**,**d**) A CD4+/CD8+ trend chart of the six time points before and after vaccination.

**Table 1 vaccines-10-00647-t001:** Characteristics of the opinion poll samples (*n* = 395).

Characteristics	*n*	%
**Gender**		
Male	170	43.04
Female	225	56.96
**Age**		
Age12–18	5	1.27
Age18–25	80	20.23
Age25–40	163	41.27
Age40–60	145	36.71
Age60–65	2	0.51
**Education**		
Bachelor’s degree	174	44.05
Bachelor’s degree below	76	19.24
Master’s degree or above	145	36.71
**Occupation**		
Student	108	27.34
Medical personnel	22	5.57
Personnel of state organs and institutions	83	21.01
Professional and technical personnel (teachers, lawyers, engineers and technicians, etc.)	73	18.48
Company employees	37	9.37
Self-employed person	15	3.8
Retirees	26	6.58
Freelancer	23	5.82
Others	8	2.03
**Health**		
Good	348	88.1
General	43	10.89
Poor	4	1.01
**Basic illness**		
CVD	12	3.04
Chronic tumor	5	1.27
Chronic respiratory disease	11	1.78
Immune deficiency diseases	3	0.76
Other chronic diseases	21	5.32
COVID-19	0	0
**Vaccination of booster COVID-19 Vaccine**	138	34.94

## Data Availability

The data supporting the findings of this study are available on reasonable request from the corresponding author Huanjie Li.

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
