# Peer review of "Opinion Polls and Antibody Response Dynamics of Vaccination with COVID-19 Booster Vaccines"

_vaccines, 2022, doi:10.3390/vaccines10050647_

Round 1
Reviewer 1 Report
This paper presents a statistical investigation to understand the reliability and coverage limits of different vaccination programs to investigate the dynamics of herd immunity in SARS-CoV-2 pandemics. The authors consider a test over almost 400 citizens with a coverage of different ages. One of the challenging objectives consists in understanding the role of heterogeneity (in this specific case related to the age).
This topic is very important as it offers an important information both to researchers and to risk managers in charge of organizing vaccination programs. The statements within the lines 34 and 43 is correct. Indeed, the present knowledge on the effective action of vaccines is still limited. However, the authors might improve the quality of their paper by giving indications on how their studies might effectively contribute to the research activity in the field.
For instance, it is useful to modelists, see
https://www.worldscientific.com/doi/epdf/10.1142/S0218202521500524
and therein bibliography.
An other interesting field is the study of viral charge
J.D.Challenger, C.Y.~Foo, Y.~Wu, A.W.C.~Yan, M.M.~Marjaneh, F.~Liew, R.S.~Thwaites, L.C.~Okell and A.J.~Cunnington, Modelling upper respiratory viral load dynamics of SARS-CoV-2, BMC Medicine, 20, 25, (2022).
Otherwise, it is a good paper which merits publication, while a broader vision on future perspectives would give to this paper an improved scientific impact
Reviewer 2 Report
The manuscript ID vaccines-1680761 has been devoted to mainly present an analysis related to particular opinion polls and antibody response dynamics involved with vaccination with COVID-19 booster vaccines. A list of comments for the authors is below:
- It is not clearly described how was designed the specimen collection and the size of the samples.
- Has really any relevance the education (Table 1) for this topic? Other parameters like home office, fat mass, or time of exercise in a week, could not be more appropriate to be analyzed?
- I wonder what are the differences between general and good health that should be considered for this topic as stated in table 1? What is the reference?
- What kind of booster vaccines was considered?
- A description or citation to justify the parameters and conditions for the laboratory analysis is missing in the report.
- Please deeper explain the dispersed data depicted in figure 5.
- The conclusions should be extended to include perspectives.
- The main findings ought to be confronted with other solutions proposed to tackle the covid19 in order to highlight this research. The authors are invited to see for instance doi:10.3390/cells9122679; doi: 10.1016/j.it.2020.10.004
- It is kindly suggested to split the collective citations in order to better justify the selection of each reference for this important topic.
- If possible, it would be helpful to comment with better details about the impact of the Booster Vaccines taking into account the new mutation of the SARS-CoV-2.
Author Response
Dear Editors and Reviewers,
Thank you for sending our manuscript to two experts for peer-review. Reviewers gave professional and constructive suggestions, which will help us to improve the quality of the manuscript. We carefully checked the whole manuscript and made some additions according to suggestions as best as we can. The modifications have been highlighted as yellow in new manuscript. We hope they can meet all the requirements. We will answer the reviewers' comments and suggestions point by point.
Point 1: It is not clearly described how was designed the specimen collection and the size of the samples.
Response 1: We appreciate those suggestions and add the description of the specimen collection and the size of the samples. “We collected 2 mL blood and 2 mL serum from volunteers by venous blood collection. The time points of collection samples are 7 days before booster vaccination, 7 days after vaccination, 14 days after vaccination, 21 days after vaccination, 28 days after vaccination and 56 days after inoculation.” We have revised the section 2.3. Specimen Collection in the article.
Point 2: Has really any relevance the education (Table 1) for this topic? Other parameters like home office, fat mass, or time of exercise in a week, could not be more appropriate to be analyzed?
Response 2: Thanks for this suggestion. I regret not considering this before designing the questionnaire. The research on the COVID-19 is still ongoing, and your suggestions will be fully referred to in the subsequent investigations. We designed the education level option in the questionnaire because this questionnaire mainly examines people's awareness and willingness to boost the COVID-19 vaccine, and the concept of disease and self-care and epidemic prevention is closely related to education level. Therefore, the questionnaire of the research designed this option. The other parameters, as the reviewers mentioned, may not be closely related to vaccination intentions, but related to the changes of vaccine antibody levels. We added this part in the discussion.
Point 3: I wonder what are the differences between general and good health that should be considered for this topic as stated in table 1? What is the reference?
Response 3: In Table 1, healthy means that the participants do not have any underlying diseases, have good immunity, and rarely catch colds, while healthy generally means that the participants have some underlying diseases that do not affect their normal life. Taken the suggestion of reviewer, we added the explanations and references to the manuscript.
[23] Mahara, G., Liang, J., Zhang, Z., Ge, Q., & Zhang, J. (2021). Associated Factors of Suboptimal Health Status Among Adolescents in China: A Cross-Sectional Study. Journal of multidisciplinary healthcare, 14, 1063–1071. https://doi.org/10.2147/JMDH.S302826.
[24] Merriam-Webster. Definition of HEALTH. Dictionary. Available from: https://www.merriam-webster.com/dictionary/health. Accessed March 21, 2019.
Point 4: What kind of booster vaccines was considered?
Response 4: We apologies that we did not clearly mark the types of booster vaccines. We have made changes based on your comments. The vaccines mentioned in the article are COVID-19 inactivated virus vaccine (Sinopharm, Beijing). It has been marked in section 2.1 Trial Design and Participants.
Point 5: A description or citation to justify the parameters and conditions for the laboratory analysis is missing in the report.
Response 5: We apologize for the confusion caused by missing contents. Detailed parameters and conditions for the laboratory analysis are described in Section 2.4. “The collected blood samples were centrifuged at 3000 rpm for 10 minutes to isolate serum and saved at 4℃. Experiments were carried out in the BSL-2 laboratory. All samples are processed in a biological safety cabinets (BSC).”“The luminometer measured the relative luminescence value RLU average of calibrator 1 and calibrator 2 three times, and calculated cut-off index (COI). When the sample test value <1.0, the 2019-nCoV antibody test result was judged to be negative. When the sample test value ≥1.0, the 2019-nCoV antibody test result is determined to be positive.”
Point 6: Please deeper explain the dispersed data depicted in figure 5.
Response 6: We appreciate the reviewer taking time to point out these deficiencies. Taken this suggestion, we have revised the manuscript. Discrete analysis was described in more details about Figure 5 in Section 3.5 Correlation Analysis of SARS-CoV-2 S-specific Binding Antibodies Concentration, and we would appreciate your review again.
Point 7: The conclusions should be extended to include perspectives
Response 7: This suggestion is very constructive and we have made amendments in Section 5 based on this comments.
Point 8: The main findings ought to be confronted with other solutions proposed to tackle the covid19 in order to highlight this research. The authors are invited to see for instance doi:10.3390/cells9122679; doi: 10.1016/j.it.2020.10.004
Response 8: As the reviewer mentioned, in order to solve the problem of the new crown, we should combine the analysis from the transmission mechanism, vaccine prevention, treatment and other aspects. In response to this suggestion, we have made a corresponding discussion in the fourth part. The two papers here also give us some hints and we cite them in the manuscript (38 and 39).
[38] Harrison, A.G., T. Lin, and P. Wang, Mechanisms of SARS-CoV-2 Transmission and Pathogenesis. Trends Immunol, 2020. 41,1100-1115.[Doi:10.1016/j.it.2020.10.004]
[39] García-Pérez, B.E., et al., Taming the Autophagy as a Strategy for Treating COVID-19. Cells, 2020. 9.[Doi:10.3390/cells9122679]
Point 9: It is kindly suggested to split the collective citations in order to better justify the selection of each reference for this important topic.
Response 9: We carefully checked the whole manuscript. Citation were collated again to see if they were representative. We made appropriate adjustments and many thanks for the reviewer's suggestion.
Point 10: If possible, it would be helpful to comment with better details about the impact of the Booster Vaccines taking into account the new mutation of the SARS-CoV-2.
Response 10: The reviewer gave us an important suggestion, and we are sorry for ignoring it. We have added the discussion to the end of the manuscript about the positive effects of the virus variant. “Several reports have also demonstrated that booster vaccines induce significant increases in neutralizing antibody titers against wild-type SARS-CoV-2 virus as well as beta and delta variants [37] .”

Round 2
Reviewer 2 Report
The authors have successfully clarified most of the points raised in the review stage. The results are valuable and the conclusions worth being released to the scientific community. Then, I can recommend this report to be considered for publication in the prestigious journal Vaccines as it is.